# Lactoferrin Isolation and Hydrolysis from Red Deer (*Cervus elaphus*) Milk and the Antibacterial Activity of Deer Lactoferrin and Its Hydrolysates

**DOI:** 10.3390/foods9111711

**Published:** 2020-11-21

**Authors:** Ye Wang, Alaa El-Din A. Bekhit, Susan L. Mason, James D. Morton

**Affiliations:** 1Department of Wine, Food and Molecular Bioscience, Lincoln University, PO Box 85084, Lincoln 7674, New Zealand; sue.mason@lincoln.ac.nz (S.L.M.); james.morton@lincoln.ac.nz (J.D.M.); 2Department of Food Science, University of Otago, PO Box 56, Dunedin 9054, New Zealand; aladin.bekhit@otago.ac.nz

**Keywords:** deer milk, whey proteins, alpha-lactalbumin, beta-lactoglobulin, lactoferrin fractionation, lactoferrin hydrolysis, antibacterial activity, *E. coli* inhibition

## Abstract

Lactoferrin (Lf) and other whey proteins have been isolated from red deer milk for the first time using a three-step anion and cation exchange chromatography protocol. The separated deer Lf was subject to in vitro gastric and duodenal digestions to generate peptides. The purity of the deer Lf and its hydrolysis products were analyzed by SDS-PAGE. The antibacterial activity of the deer Lf and its hydrolysates were investigated and was compared to cow counterpart. Gastric and duodenal digested deer Lf had strong bactericidal activity against *E. coli* ATCC 25922 with minimum inhibition concentration (MIC) of 280 µM and 402 µM, respectively. These results suggest that deer milk contains bioactive whey proteins and can generate bioactive peptides, which can benefit human health by inhibiting food-borne pathogenic bacteria.

## 1. Introduction

New Zealand has a large population of farmed deer. Therefore, it is economically viable to fully exploit co-products generated from the deer industry and promote new deer products on the market. Based on the large population of mated hinds in New Zealand, deer milk is a commercial possibility that could add another income stream to the deer industry and stabilize deer farming against the volatility of fluctuating venison prices and demand for deer velvet. Our research group has been investigating deer milk for several years that was focused on understanding the general characteristics of deer milk and the protein composition, and we produced the world’s first cheese made from red deer milk [1]. Depending on the method of coagulation, there are two types of whey: sweet whey, which is obtained by the coagulation of using casein rennet, as in cheese making, and acid whey, which is obtained by casein acid precipitation at pH 4.6 at room temperature [2]. Whey is a co-product of cheese making and contains valuable nutrients such as protein, lactose, and minerals. These can be recovered and utilized in various products. Historically, deer milk has been used not only as a food but also for medicinal purpose [3], which encourages the use of any co-products generated from deer milk such as whey. The nutrient compositions vary among deer subspecies. For the Cervinae subfamily, the ranges of total proteins in red and fallow deer are 6.8–10.6 g/100 g and 6.5–8.0 g/100 mL, respectively [4]. Casein accounts for the majority of proteins in the milk from red deer, fallow deer, and roe deer. Fat content among these species also has a wide variation, especially in red deer milk, being from 7% to 19.7%. The lactose contents in the milk of reindeer and white-tailed deer are 2.5–3.7% and 2.7%, respectively, which is lower than in the milk of other deer subspecies. Total protein in deer milk is more than twice that of bovine milk. Similar to bovine milk, the major protein in deer milk is casein [4].

Most research was focused on whey proteins from cattle. Cow lactoferrin (Lf) has 708 amino acids with molecular weight of 78.1 kDa [5]. Cow β-Lactoglobulin (β-Lg) is a small, soluble, globular protein, containing 162 amino acids in a single peptide chain with a molecular weight of 18.3 kDa. β-Lg is the major whey protein found in milk from cow, sheep, goat and other ruminants. Lf is an iron binding glycoprotein of the transferrin family [6]. Limited information is available on the content and composition of deer whey proteins. Wang et al. (2017) reviewed the composition and protein concentrations of deer milk [4]. The Lf content in deer milk is 0.26 mg/mL, which is higher than that in cow milk, 0.233 mg/mL [7]. The major whey protein in deer milk is β-Lg 0.978 mg/mL [8].

Whey proteins and hydrolysates from milk of various species exhibit significant antimicrobial activities. Lf has been shown to inhibit the growth of a number of pathogenic Gram-positive and Gram-negative bacteria (including antibiotic-resistant strains), fungi, parasites and even viruses in both in vitro and in vivo studies [9,10,11]. Many biologically active peptides are latent in the Lf sequence and can be released by proteolysis during gastrointestinal digestion or by in vitro hydrolysis processes. A great number of antimicrobial peptides have been identified within the sequences of Lf from milk, most of which have been reported in cow milk. The MilkAMP database contains over 300 Lf peptides that have been investigated for antimicrobial activities. Dozens of peptides derived from Lf have exhibited stronger and broader spectrum activity against bacteria, fungi, parasite, virus and yeast than Lf itself [12].

The aim of the present study was to determine the bioactivities of individual whey proteins and hydrolysates from deer milk. To achieve this goal, lactoferrin and other whey proteins were isolated from deer milk using ion exchange chromatography and hydrolysates from deer Lf were generated using in vitro gastric and duodenal digestions.

## 2. Materials and Methods

### 2.1. Materials

Ion exchange media and columns were purchased from GE Healthcare, Sweden. All chemicals for SDS-PAGE were obtained from Bio-Rad (Berkeley, California, USA), unless otherwise stated. Commercial cow Lf was a generous gift from Dr. Alaa El-Din Ahmed Bekhit, University of Otago. The following enzymes were used for the in vitro digestions: pepsin from porcine stomach mucosa (3200–4500 units/mg protein, Sigma-Aldrich, Auckland, New Zealand), trypsin from bovine pancreas (8300 units/mg protein, Sigma-Aldrich, Auckland, New Zealand) and chymotrypsin from bovine pancreas (40 units/mg protein, Sigma-Aldrich, Auckland, New Zealand). *Escherichia coli* ATCC 25922, *Staphylococcus aureus* ATCC 25923 and *Lactobacillus acidophilus* ATCC 4356 from Institute of Environmental Science and Research (ESR, Christchurch, New Zealand) were used in antibacterial activity assays. The generated hydrolysates were used to investigate their modulating activity against *Escherichia coli* ATCC 25922, *Staphylococcus aureus* ATCC 25923 and *Lactobacillus acidophilus* ATCC 4356. *E. coli* ATCC 25922 was chosen because it is a Clinical & Laboratory Standards Institute (CLSI) control strain for antimicrobial susceptibility testing. This organism does not produce verotoxin. *S. aureus* ATCC 25923 is the recommended reference strains for antibiotic susceptibility testing. Hence, proteins and hydrolysates from deer and bovine whey were tested on these two strains for antibacterial activities. *L. acidophilus* is a part of the natural microflora of the human intestinal tract. It is added into dairy products for its potential probiotic effects. *L. acidophilus* was chosen to test the effects of milk whey proteins and hydrolysates on human gut microbiota.

### 2.2. Fast Protein Liquid Chromatography

This study used unpasteurized frozen deer whole milk and cow whole milk (control group) from Canterbury deer and cow farms (New Zealand). The same processes were applied on both deer and cow milk. The milk was centrifuged (Heraeus^TM^ Multifuge^TM^ X3R centrifuge, Thermo Scientific, Dreieich, Germany) at 10,000× *g* for 30 min at 4 °C to remove the cream fraction. Sweet whey was prepared by adding rennet (Renco, Eltham, New Zealand) at 2 mL/L into skimmed milk and incubated for 60 min at 37 °C in a shaking water bath; then, it was centrifuged at 10,000× *g* for 30 min at 4 °C. The supernatant (deer milk whey and cow milk whey) was decanted, aliquoted and stored at −80 °C until required for analysis.

The diagram of whey protein fractionation from deer milk is shown in Figure 1. The deer whey proteins were fractionated using cation exchange chromatography following the method reported by [13]. The whey was pre-filtered (DVPP, 0.45 µm, Durapore^®^, Millipore, Carrigtwohill, Ireland) and the pH was adjusted to 3.8 with 1 M HCl and initially subjected to anion exchange chromatography using two × 5 mL HiTrap Q-FF cartridges linked in series. The columns were equilibrated with 20 mM sodium citrate, pH 3.8, containing 40 mM NaCl, and were installed in a BioLogic DuoFlow FPLC (GE Healthcare, Auckland, New Zealand). Bound protein was eluted from the HiTrap Q-FF with two column volumes of 20 mM sodium citrate buffer, pH 3.8, containing 1 M NaCl (Step 1). The unbound protein fraction was collected and adjusted to pH 7.0 and loaded onto five × 5 mL HiTrap SP-FF cartridges linked in series (Step 2). The columns were equilibrated in sodium phosphate (20 mM, pH 7). Bound proteins were eluted stepwise with two column volumes each of phosphate buffer containing (i) 0.1 M NaCl, (ii) 0.4 M NaCl and (iii) 1.0 M NaCl. The unbound proteins from step 2 were adjusted to pH 6.5 with 1 M phosphoric acid, and were loaded onto two × 5 mL HiTrap Q-FF columns equilibrated with 20 mM sodium phosphate, pH 6.5 (Step 3). The bound protein was eluted with a gradient (0–50%) of 1 M NaCl in 20 mM sodium phosphate, pH 8.0, over 60 min. Proteins were collected and pooled corresponding to the peaks in the chromatograms and concentrated and desalted using Vivaspin^®^ 20, with 10 kDa molecular weight cut off centrifuge filter units (Sartorius, Epsom, United Kingdom). Commercial cow Lf was used for in vitro digestion and antibacterial activity assay.

### 2.3. In Vitro Digestion of Lf

In vitro digestions were performed in two steps, representing the gastric and duodenal processes [14]. In the first gastric digestion step, the pH of protein samples was adjusted to 2.5 with 0.5 M HCl. Pepsin was dissolved in Simulated Gastric Fluid (SGF) (0.15 M NaCl, pH 2.5), and was added to the protein samples to give a final concentration of 165 U of pepsin per mg of protein. Half of the gastric digested sample was further hydrolyzed in duodenal digestion. The other half of the gastric digested sample was centrifuged at 15,000× *g* for 30 min at 4 °C and the supernatant was collected and stored at −20 °C for antibacterial activity assay.

In stage two duodenal digestion, the pH of the digesta from the gastric digestion was firstly adjusted to 6.5 with 0.1 M NaOH. Trypsin 34.5 U per mg of whey protein and chymotrypsin 0.4 U per mg of whey protein were added into the digesta. The digestion was performed in a shaking incubator (170 rpm) at 37 °C, 60 min for stage one and 30 min for stage two. Aliquots were withdrawn from the mixture at the beginning and the end of gastric and duodenal digestions to evaluate protein hydrolysis profile using SDS-PAGE. Hydrolysis was terminated by heating at 80 °C for 15 min. All the digested protein samples were centrifuged at 15,000× *g* for 30 min at 4 °C and the supernatant was collected and stored at −20 °C for antibacterial activity assay.

### 2.4. OPA Assay for Lf Peptide Production

The o-Phthaldialdehyde (OPA) assay was performed to assess the peptides production from proteolysis of whey proteins in buffered solutions. Fresh OPA reagent was prepared as follows: 25 mL of 100 mM sodium tetraborate was added to 2.5 mL of 20% (*w*/*w*) SDS; 40 mg of OPA dissolved in 1 mL methanol; and 100 µL of β-mercaptoethanol, and the final volume was made up to 50 mL using distilled water. Aliquots from hydrolysis were incubated with 0.75 M Trichloroacetic acid (TCA) at a sample:TCA ratio of 1:3 at 4 °C and then centrifuged (4000× *g* for 5 min) to eliminate any interference due to undigested protein. The assay mixture contained the supernatant of hydrolyzed samples (50 µL) and 1.0 mL of OPA reagent, was mixed briefly by inversion and incubated for 2 min at room temperature. The absorbance of the mixture was measured at 340 nm using a spectrophotometer (V-1200, VWR^®^, Atlanta, GA, USA). Leucine (Leu) was used for the construction of a standard curve. The peptide production was expressed as amount of peptide produced per µg of original protein (µg Leu equivalent/µg protein) to make all proteins with different starting amount comparable.

### 2.5. SDS-PAGE

The purity of deer milk whey proteins isolated from the FPLC and the whey protein digestibility were analyzed by SDS-PAGE on a 12% Bis-Tris electrophoresis gel run in a Bio-Rad mini-gel electrophoresis system. All chemicals were from (Bio-Rad, Auckland, New Zealand). SDS-PAGE was applied according to a standard protocol (Laemmli 1970). Twelve percentage Bis-Tris gels were made with resolving gel 1.5 M Tris (pH 8.8), stacking gel 0.5 M Tris (pH 6.8), 30% Acrylamide/Bis (37.5:1, 2.6% C), 10% SDS solution, 10% APS and TEMED. Bio-Rad Precision Plus ProteinTM standard (#1610373) was used as a molecular weight marker. Protein samples were diluted to 1.11 g/L with 4× sample buffer (6 μL), which contained 0.23 M Tris (pH 6.8), 8% SDS, 40% Glycerol, 0.08% Bromophenol Blue and 4% Mercaptoethanol. The volume was adjusted to 24 μL with water and heated at 72 °C for 10 min. The samples (24 μL) and 10 μL of the molecular marker were loaded into the wells on Bis Tris gel. Electrophoresis was performed at a constant voltage of 120 v for 90 min at 18 °C room temperature. Gels were fixed in 50% methanol and 7% acetic acid for 15 min with rocking and then washed three times for 15 min with RO water. Gels were stained with GelCodeTM Blue (#24590, Thermo Scientific, Dreieich, Germany) for 1 h at 18 °C room temperature and then destained overnight with continuous shaking in RO water at 4 °C. Stained SDS gels were scanned and saved digitally with a CS9000F Mark II scanner (Canon New Zealand, Christchurch, New Zealand).

### 2.6. Bacterial Strains

The antibacterial activity assay for deer milk whey proteins and hydrolysates was performed according to the European committee for antimicrobial susceptibility testing protocol [15]. Working cultures were made from culturing bacteria that were kept on an agar plate at 4 °C. A colony of bacteria was picked up from working culture and enumerated in broth at 37 °C overnight. Bacteria in nutrient/MRS broth were serially diluted with broth to 5 × 10^5^ cell/mL for *E. coli* ATCC 25922 and *S. aureus* ATCC 25923, 1.5 × 10^8^ cell/mL for *L. acidophilus* ATCC 4356. This was because *L. acidophilus* ATCC 4356 with lower cell density (10^5^ cell/mL) could not achieve a good growth curve. In total, 100 μL of the diluted bacteria was transferred into each well of a 96-well plate, mixed with 100 μL protein or hydrolysate in different concentrations (0.125–4.0 g/L). The 96-well plates were incubated at 37 °C and the OD600 was measured hourly by plate reader for 24 h for *E. coli* ATCC 25922 and *S. aureus* and after 48 h incubation for *L. acidophilus* ATCC 4356 with shaking for 60 s before reading. Minimum inhibitory concentration (MIC), defined as the lowest concentration of sample that caused complete inhibition of bacterial growth, was determined during the 24/48-h incubation. Minimum bactericidal concentration (MBC), defined as the concentration at which there was ≥99.9% decrease in viable cells, was determined by taking 100 μL from wells where no growth was detected, which was then spread on agar plates and incubated at 37 °C for 48 h for bacterial viable count [16].

Positive antibiotics control group was bacteria in 100 μL broth + 100 μL Penicillin (10,000 units/mL) and Streptomycin (10,000 µg/mL) (GIBCO 15140, Invitrogen^TM^, Auckland, New Zealand). Negative broth control group was bacteria in 200 μL broth. To test for sterility, a further control was uninoculated broth (200 μL). All assays were performed in triplicate.

### 2.7. Determination of Antibacterial Activity

For the bacterial growth evaluation, the generation time was calculated to define the bacterial growth time per generation:$$G=\frac{t}{3.3\;log\frac{b}{B}}$$
where *G* is the Generation time, *t* is the time interval in minutes, *B* is the number of bacteria at the beginning of a time interval and *b* is the number of bacteria at the end of the time interval.

### 2.8. Statistical Analysis

The data were subjected to a one-way analysis of variance (ANOVA), followed by the Sidak correction in the General Linear Model to determine the significant differences between samples and control groups, and intergroup comparisons at *p* < 0.05 level using Minitab (Minitab Inc., version 17, Sydney, NSW, Australia). All triplicate independent experiments were carried out and the antibacterial activity assays were carried out in triplicate (n = 3), and reported values are mean ± standard deviation.

## 3. Results

### 3.1. Fractionation of Whey Proteins from Deer Milk

The chromatograms of FPLC step 1, 2 and 3 are shown in Figure 1. In Figure 1A, a protein fraction putative glycocaseinomacropeptie (GMP) (No. 53 and 54) was fractionated. It can be concluded that at pH 3.8, all deer whey proteins, except for the isolated protein in this step, should carry a positive charge, and therefore, did not bind to the anion exchange Q-FF resin. The isolated protein from deer milk is negatively charged, and bound to the anion exchange media at pH 3.8 and was eluted by 1 M NaCl in 20 mM sodium citrate buffer. Further investigation will be carried out to identify the fractionated protein in this step.

Whey proteins from deer milk that were weakly bound to the cation exchanger SP-FF were eluted by the sodium phosphate buffer containing 0.1 M NaCl. Bound proteins were eluted stepwise with 20 mM sodium phosphate buffer, pH 7.0, containing 0.4 M NaCl and then 1.0 M NaCl, which, respectively, corresponds to the second and third peaks in Figure 1B. The proteins were the putative deer immunoglobulin (Ig) and lactoperoxidase (Lp), lane 5 in Figure 2, and the putative Lf, lane 6 in Figure 2. This result manifested that at pH 7, deer Ig, Lp and Lf were positively charged and bound to the cation media. The identity of deer Lf was confirmed by analyzing the amino acid sequence from mRNA and comparison with an in-house deer protein sequence database in the Department of Biochemistry, University of Otago (AP. Alan Carne). Isolated deer Lf was used for in vitro digestion and antibacterial activity assay.

The unbound proteins from cation exchanger in FPLC step 2 contained mainly α-La and β-Lg (lane 7 Figure 2). At pH 6.5, deer α-La and β-Lg were negatively charged and bound to the anion exchange resin. Figure 1C shows that the peaks of putative deer α-La and β-Lg were overlapped with each other. The result was verified by SDS-PAGE that in the former part (first half) of eluted protein, there were both deer α-La and β-Lg (lane 8 Figure 2). However, with the increase of salt concentration and pH, the latter part (second half) of eluted protein was mainly deer β-Lg (lane 9 Figure 2).

### 3.2. Digestibility of Deer and Cow Lf

The digestibility of deer and cow Lf was determined by simulated gastric and duodenal digestion (Figure 3). Peptide production from deer and cow Lf continued during the 60 min gastric and 30 min duodenal digestions. Digestion of deer Lf generated over twice the number of peptides (22.17, 63.57 µg Leu equivalent/µg protein) produced by cow Lf (8.73, 25.73 µg Leu equivalent/µg protein). This suggested that deer Lf was more susceptible to pepsin, trypsin and chymotrypsin hydrolytic activities. Both deer and cow Lf were susceptible to pepsin and were largely hydrolyzed during the gastric digestion stage (Figure 3, lane 2 and 5). A band with a molecular weight around 20 kDa remained after 60 min of gastric digestion of deer Lf, which was further hydrolyzed in the duodenal digestion. However, there was still a trace of this protein after duodenal digestion (Figure 3, lane 3). There was no trace of cow Lf after 30 min of duodenal digestion (Figure 3, lane 6).

### 3.3. Antibacterial Activities of Deer and Cow Lf and Their Hydrolysates

#### 3.3.1. Antibacterial Activity of Deer and Cow Lf

Growth curves of *E. coli* ATCC 25922, *S. aureus* ATCC 25923, and *L. acidophilus* ATCC 4356 in the presence of different concentrations of deer Lf, cow Lf and their hydrolysates were investigated. Neither deer Lf nor cow Lf achieved MIC or MBC against *E. coli* ATCC 25922 (Figure 4 and Figure 5), *S. aureus* ATCC 25923 and *L. acidophilus* ATCC 4356 (within the tested concentrations from 0.125 to 4 g/L). However, treatment with cow and deer Lf significantly increased (*p* < 0.05) the generation times of these bacteria compared with broth control group (Table 1). The generation time of *E. coli* ATCC 25922, *S. aureus* ATCC 25923 and *L. acidophilus* ATCC 4356 in the presence of cow Lf was significantly longer (*p* < 0.05) than that of deer Lf at the same concentration (Table 1). This indicated that, as an intact protein, both deer and cow Lf can slow the growth of *E. coli* ATCC 25922, *S. aureus* ATCC 25923 and *L. acidophilus* ATCC 4356. Cow Lf showed stronger ability than deer Lf at the same concentration.

#### 3.3.2. Antibacterial Activity of Deer and Cow Lf Hydrolysates

The growth curves of *E. coli* ATCC 25922 treated with different concentrations of deer and cow Lf hydrolysates from in vitro gastric and duodenal digestions are shown in Figure 4 and Figure 5. Deer Lf gastric hydrolysates reached the MIC at 280 µM, which is also the MBC against *E. coli* ATCC 25922 (Table 1). Duodenal hydrolysates from deer Lf at the concentration of 402 µM was both the MIC and the MBC against *E. coli* ATCC 25922 (Table 1). This showed that deer Lf hydrolysates from both gastric and duodenal digestions had strong bactericidal activity. The cow Lf gastric and duodenal hydrolysates tested at high concentration (360 and 530 µM) did not achieve either MIC or MBC, which suggests that deer Lf hydrolysates from gastric and duodenal digestions exhibited stronger bactericidal activity against *E. coli* ATCC 25922 than the cow Lf hydrolysates at the same concentration. For *S. aureus* ATCC 25923, bacteria treated with deer Lf gastric hydrolysates (560 µM) and duodenal hydrolysates (804 µM) and cow Lf gastric (360 µM) and duodenal hydrolysates (530 and 1060 µM) had significantly longer (*p* < 0.05) lag time and generation time than the nutrient broth control group (Table 1), which suggests that higher peptide concentration resulted in slower bacterial growth.

## 4. Discussion

The present study successfully isolated Lf and β-Lg from deer milk by using cation and anion exchange chromatography, respectively. Lf from several different species (sheep, goat, camel, alpaca, elephant and human) can be eluted by 20 mM sodium phosphate containing high concentration NaCl (1M) at a neutral pH [16]. Deer Lf, similar to the Lf from other species, is positively charged and strongly bound to cationic resin at pH 7, and can be eluted by 1 M NaCl in 20 mM sodium phosphate buffer. Lf, Lp, serum albumin, β-Lg and α-La from deer milk have similar molecular weights to those from cow milk, which are Lf (78 kDa), Lp (70 kDa), BSA (69 kDa), β-Lg (18.4 kDa) and α-La (14.2 kDa) [17].

This study attempted to separate deer α-La and β-Lg in FPLC using Q-FF anion exchanger. These two proteins were eluted with a linear gradient from 0–50% of 1 M NaCl in 20 mM sodium phosphate buffer. Fractionation of β-Lg was achieved with no contaminating proteins in latter part of eluent, whereas the former part of eluted α-La had a trace amount of β-Lg. The present results showed the difficulty of recovering individual α-La with high purity within one step in FPLC. El-Sayed and Chase (2010) studied whey proteins fractionation and published several papers on purification of whey α-La and β-Lg using cation exchange chromatography. They show a novel consecutive two-stage separation process was developed to separate α-La and β-Lg from whey concentrate mixtures [18]. SP-FF column with 0.1 M sodium acetate buffer for column equilibration at pH 3.7 and 0.1 M Tris-HCl for elution at pH 5.2 and pH 7 resulted in high purity and recovery of α-La and β-Lg. This may provide a good solution of α-La and β-Lg fractionation with high purity from deer milk in the future work by using cation exchange chromatography rather than anion exchanger.

In the present study, simulated gastric and duodenal digestions were applied to hydrolyze deer and cow Lf based on mimicking conditions found in human gastrointestinal tract. Results from present study showed both deer and cow Lf were susceptible to pepsin, trypsin and chymotrypsin. Deer Lf produced more peptides than cow Lf during both gastric and duodenal digestions. This indicated that deer Lf is more susceptible to pepsin, trypsin and chymotrypsin than cow Lf. This further implied that there might be different amino acid sequences in deer Lf, which resulted in more enzymatic cleavage sites, and therefore, produced more hydrolysates than cow Lf.

Lf is a precursor of diverse bioactive peptides which can be released by enzymatic proteolysis. Currently, over 300 peptides derived from Lf have been studied for their antimicrobial activity and are collected in the MilkAMP database [19]. *E. coli* ATCC 25922 and *S. aureus* ATCC 25923 were chosen as representatives of pathogenic and gut microbiota. This research found that the lag time and generation time of *E. coli* ATCC 25922 and *S. aureus* ATCC 25923 were significantly increased (*p* < 0.05) when treated with deer and cow Lf, which means deer and cow Lf had slowed bacterial growth but not bacteriostatic. In vitro studies have revealed that Lf exhibited both a bacteriostatic and bactericidal effect against different strains of *E. coli*, such as enteroaggregative *E. coli* (EAEC), enterohemorrhagic *E. coli* (EHEC) and enteropathogenic *E. coli* (EPEC) [20]. The MIC of Lf against *E. coli* varies among different *E. coli* strains and animal species from 0.1 g/L to 10 g/L, which resulted from different virulence of different *E. coli* strains [21,22,23,24]. The MICs of Lf against *S. aureus* differed in strains from 0.5 g/L against *S. aureus* JCM 2151 [25] to 8 g/L against *S. aureus* ATCC 25923 after 24 h of incubation [26]. Chen et al. (2013) incubated *S. aureus* ATCC 25923 in tryptic soy broth at 1.5 × 10^8^ cfu/mL in antibacterial activity experiments. The present study incubated *S. aureus* ATCC 25923 in nutrient broth at 3–5 × 10^5^ cell/mL based on EUCAST guidelines [15]. Mathematical models have predicted that the antibiotic treatment can be affected by cell density and treatment protocols based on conventional (density-independent) MICs can fail to clear higher density infections [27]. This indicated the MIC against *S. aureus* can be different at different bacterial cell density. 

In the present study, pooled Lf peptides from gastric and duodenal digestions were tested on *E. coli* ATCC 25922 and *S. aureus* ATCC 25923. Deer Lf hydrolysates had stronger antibacterial activity than cow Lf hydrolysates against *E. coli* ATCC 25922 growth. Deer Lf gastric digested hydrolysates at 280 µM (Leu equivalent) and duodenal digested hydrolysates at 402 µM (Leu equivalent) exhibited bacteriostatic and bactericidal activities against *E. coli* ATCC 25922. Pooled Lf gastric and duodenal digested hydrolysates could contain antibacterial peptides, e.g., Lf f(1–16)-(17–48) [28], Lf f(17–41) [29], Lf f(265–296) [30] and so on, which resulted in growth inhibition of *E. coli*.

## 5. Conclusions

This study provides new knowledge on deer milk whey protein isolation, hydrolysis and antibacterial activity as compared to cow milk. The results indicate that deer milk contains bioactive proteins and can generate bioactive peptides, which can benefit human health in inhibiting food-borne pathogenic bacteria *E. coli* ATCC 25922 at 280 µM (MIC) gastric hydrolysates and 402 µM (MIC) duodenal hydrolysates. The differences in deer and cow milk whey protein degradation and the ability to inhibit bacterial growth suggest that there might be different amino acids in deer Lf compared with cow Lf. The next step of our research is to determine the amino acid sequence of deer Lf and test the antibacterial activity of specific peptides encrypted in deer Lf.

## Figures and Tables

**Figure 1 foods-09-01711-f001:**
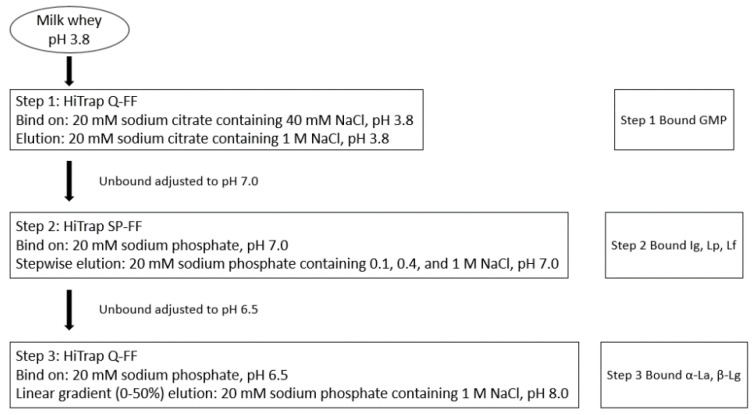
Chromatogram of Lf and β-Lg isolation from red deer milk using a three-step FPLC method. (**A**) First anion exchange chromatography to separate GMP, eluted with 20 mM sodium citrate, 1 M NaCl, pH 3.8. (**B**) Second cation exchange chromatography to separate Ig and Lp, eluted with 20 mM sodium phosphate, 0.4 M NaCl, pH 7; Lf, eluted with 20 mM sodium phosphate, 1 M NaCl, pH 7. (**C**) Third anion exchange chromatography to separate α-La and β-Lg with a linear gradient of 0–50% 20 mM sodium phosphate, 1 M NaCl, pH 8.

**Figure 2 foods-09-01711-f002:**
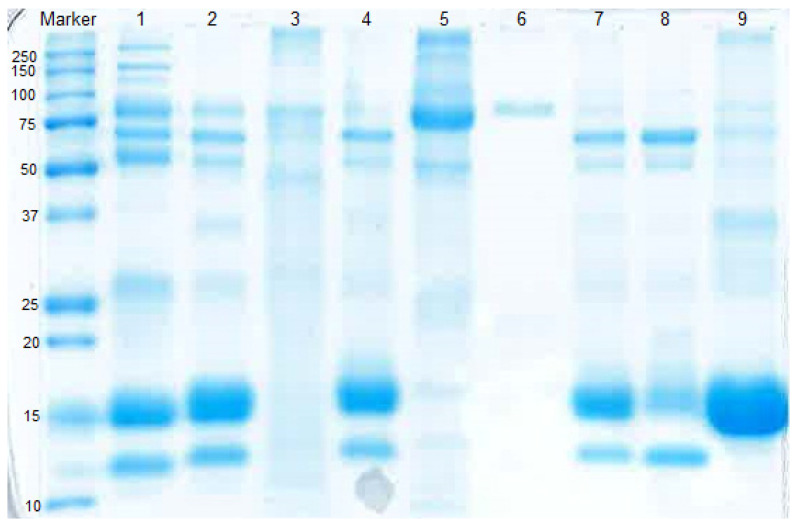
SDS-PAGE of cow and deer milk whey proteins. Lane 1: cow milk whey; lane 2: deer milk whey. Lane 3–9 are deer milk whey protein fractions from FPLC—lane 3: putative GMP from FPLC Step 1; lane 4: flow-through from FPLC Step 1; lane 5: putative Ig and Lp from FPLC Step 2; lane 6: putative Lf from FPLC Step 2; lane 7: flow-through from FPLC Step 2; lane 8: pooled fraction 31–42 (former part of eluted proteins) from FPLC Step 3; lane 9: pooled fraction 43–50 (latter part of eluted proteins) from FPLC Step 3.

**Figure 3 foods-09-01711-f003:**
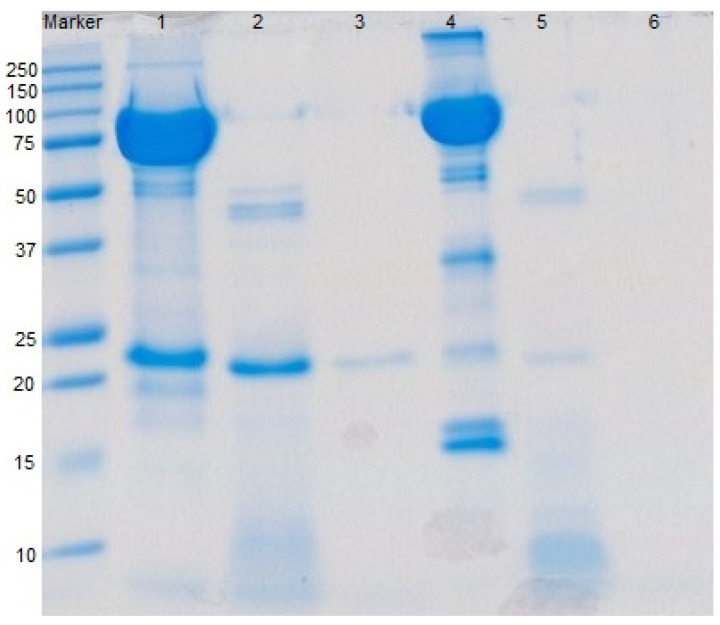
Deer and cow Lf degradation during in vitro digestion. Lane 1: deer Lf; lane 2: 60 min gastric digested deer Lf; lane 3: 30 min duodenal digested deer Lf; lane 4: cow Lf; lane 5: 60 min gastric digested cow Lf; lane 6: 30 min duodenal digested cow Lf.

**Figure 4 foods-09-01711-f004:**
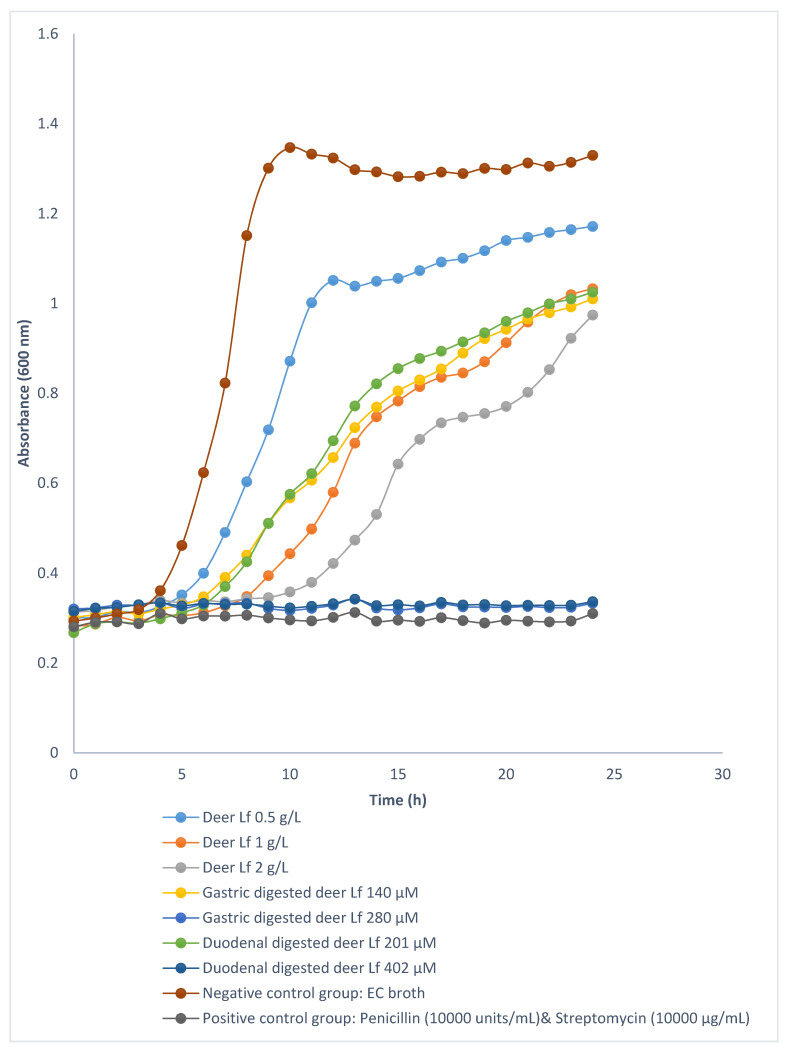
Growth curves of *E. coli* ATCC 25922 incubated with different concentrations of deer Lf and its hydrolysates at 37 °C during a 24-h incubation period. All the curves are the average of triplicate incubations.

**Figure 5 foods-09-01711-f005:**
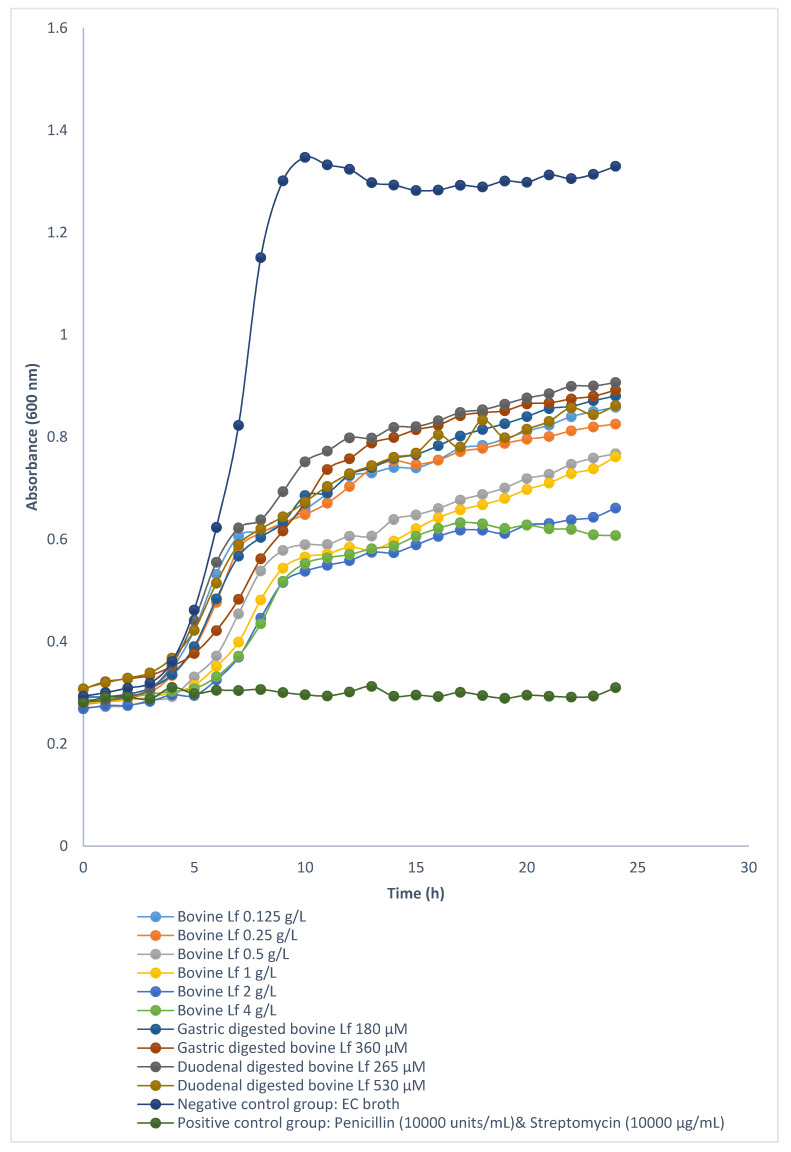
Growth curves of *E. coli* ATCC 25922 incubated with different concentrations of cow Lf and its hydrolysates at 37 °C during a 24-h incubation period. All the curves are the average of triplicate incubations.

**Table 1 foods-09-01711-t001:** Growth curve parameters of bacteria treated with deer and cow Lf and their hydrolysates.

	Concentration	*Escherichia coli* ATCC 25922		*Staphylococcus aureus* ATCC 25923		*Lactobacillus acidophilus* ATCC 4356
		Lag Time (h)	Generation Time (min)	Lag Time (h)	Generation Time (min)	Generation Time (h)
Deer Lf (g/L)	0.125	*3.00 ± 0.00*	* 34.12 ± 0.01 ^c^	4.67 ± 0.33	37.57 ± 0.09	NA
	0.25	*3.33 ± 0.33*	* 34.00 ± 0.07 ^d^	5.00 ± 0.00	37.70 ± 0.01 ^k^	10.43 ± 0.07
	0.5	** 4.33 ± 0.33*	* 33.55 ± 0.02 ^e^	5.00 ± 0.00 ^h^	37.78 ± 0.01 ^l^	9.62 ± 0.20
	1	** 7.00 ± 0.00 ^a^*	* 34.83 ± 0.10 ^f^	5.00 ± 0.00 ^i^	37.29 ± 0.02 ^m^	* 13.28 ± 0.87
	2	** 10.33 ± 0.33 ^b^*	* 34.68 ± 0.08 ^g^	5.33 ± 0.33 ^j^	37.07 ± 0.12 ^n^	12.50 ± 2.04 ^o^
Deer Lf hydrolysates (µM)	140 gastric	* 5.67 ± 0.33	* 35.24 ± 0.06	NA	NA	NA
	280 gastric	MIC	MIC	5.00 ± 0.00	37.11 ± 0.09	NA
	560 gastric	NA	NA	* 8.33 ± 0.33	37.99 ± 0.11	NA
	201 duodenal	* 4.67 ± 0.33	* 35.47 ± 0.01	NA	NA	NA
	402 duodenal	MIC	MIC	5.00 ± 0.00	36.99 ± 0.17	NA
	804 duodenal	NA	NA	* 8.00 ± 0.00	37.62 ± 0.05	NA
Cow Lf (g/L)	0.125	3.00 ± 0.00	* 35.27 ± 0.04 ^c^	NA	NA	NA
	0.25	3.00 ± 0.00	* 35.25 ± 0.01 ^d^	5.00 ± 0.00	* 38.40 ± 0.01 ^k^	12.33 ± 1.00
	0.5	4.00 ± 0.00	* 35.59 ± 0.02 ^e^	* 6.33 ± 0.33 ^h^	* 38.83 ± 0.09 ^l^	* 12.99 ± 0.29
	1	* 4.67 ± 0.33 ^a^	* 35.77 ± 0.01 ^f^	* 6.67 ± 0.33 ^i^	* 38.76 ± 0.06 ^m^	* 17.19 ± 6.60
	2	* 5.00 ± 0.00 ^b^	* 35.95 ± 0.01 ^g^	* 7.33 ± 0.33 ^j^	* 38.38 ± 0.10 ^n^	* 17.04 ± 0.91 ^o^
	4	* 4.67 ± 0.33	* 35.86 ± 0.01	* 9.33 ± 0.33	* 38.17 ± 0.02	* 15.72 ± 0.35
Cow Lf hydrolysates (µM)	180 gastric	3.33 ± 0.33	* 35.11 ± 0.09	NA	NA	NA
	360 gastric	3.33 ± 0.33	* 35.23 ± 0.16	* 11.00 ± 0.00	* 38.93 ± 0.28	NA
	720 gastric	NA	NA	5.33 ± 2.33	38.00 ± 0.22	NA
	265 duodenal	3.00 ± 0.00	* 34.80 ± 0.05	NA	NA	NA
	530 duodenal	3.00 ± 0.00	* 35.16 ± 0.12	* 7.00 ± 0.00	* 38.77 ± 0.03	NA
	1060 duodenal	NA	NA	* 8.33 ± 0.33	* 38.29 ± 0.02	NA
Broth control group		3.00 ± 0.00	32.01 ± 0.01	4.00 ± 0.00	37.46 ± 0.01	9.38 ± 0.17

The hydrolysate concentration was expressed as µM Leu equivalent. * Significantly different (*p* < 0.05) from the broth control group. a–o, *p* < 0.05 pairwise comparison within the same concentration between deer and bovine Lf. Values are means ± SD based on three observations. NA: not available.

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
