# Peer review of "Lactoferrin Isolation and Hydrolysis from Red Deer (Cervus elaphus) Milk and the Antibacterial Activity of Deer Lactoferrin and Its Hydrolysates"

_foods, 2020, doi:10.3390/foods9111711_

Round 1
Reviewer 1 Report
General comments:
The purpose of this manuscript is to isolate lactoferrin from red deer and to evaluate the antimicrobial properties of the native lactoferrin recovered as well as its hydrolysates generated after gastric and duodenal digestions. while this research is orginal and give new results, in my opinion, this masuscript has to be largely review in order to improve its quality.
Specific comments:
Title: the title is "Lactoferrin and hydrolysates from red deer...". Hydrolysates obtained from lactoferrin ou whey proteins? It is necessary to detail this point in the title by indicating that lactoferrin hydrolsayes were also studied.
Line 9: please modifiy "Correspondence:Correspondence". The corresponding author must use its institutional email and not a "gmail".
Introduction:
Lines 41 to 43. The values of protein content given by authors are dependent of cow species, seasonal changes, etc and consequently, are not always the same. Consequently, authors must provide a range of protein content and not one value.
Materials:
Line 69: Globally, for each material/chemical, please indicate the company, city, and country
Line 70: please replace "were" by "were purchased"
Line 79: the authors used unpasteurized frozen deer whole milk. How authors manage the initial bacterial contamination in milk? How did they distinguish peptides originating from gastric duodenal and peptides initially present in milk due to proteolysis by bacteria. Similarly, generally, the determination of antimicrobial activities were performed on sterile raw materials (sterilized by heat or membrane filtration) which are subsequently contaminated by different bacterial strains. In this context, how authors deal with the fact that unpasteurized milk was used?
Line 82: please replace ml/L by mL/L and make this modification throughout the manuscript. Please replace "defatted" by "skimmed"
Line 84: Authors mentioned that cow milk was used. I suppose that cow milk is a control. However, no information are given regarding the different processes applied on milk to obtained cow milk whey. Please add details about this point.
Line 123: How authors discriminate proteolysis induced by digestion vs proteolysis in initial milk due to bacterial contamination since milk was not pasteurized?
Line 134 to 141: A lot of details are missing regarding the preparation of SDS-PAGE (boiling step, add of beta mercaptoethanol, etc.).
Line 142: Similar to my previous comments: How samples were sterilized for antimicrobial analysis?
Results
Line 179: Not necessary
Line 196: The figure is not necessary. If authors decide to include this figure to the manuscript, the quality needs to be largely improved.
Figures: Globally, the graphic quality of all figures integrated in this manuscript have to be largely improved
Figure 2. A lot of detail are missing in the description of results related to Figure 2. Authors have to describe and characterize the different bands obtained in the gel. Moreover, where are the standards (b-lactoglobulin, alpha-lactalbumin)? Protein standards are crucial for protein identification. I have similar comments for Figure 3.
Table 1. I'm wondering why authors compared deer and bovine milk. Why they did not focus the results on deer milk since it is the main research mentioned in the title and abstract. Moreover, Table is quite difficult to understand: difficult in terms of statistical analysis, what is the significance of the "X" in the Table. Why the authors have chosen a pairwise comparison for the data?
Figures 4 and 5: Bad quality, overloaded figures. again, why is it necessary to compare bovine and deer milk?
Discussion
In my opinion, the discussion has to be largely improved in terms of scientific content, structuration and quality of English
Author Response
The authors thank the reviewers for their useful comments and for their time reviewing the manuscript.
Response to Reviewer 1 comments:
1. Title: has been changed to indicate both deer Lf and its hydrolysates were studied.
2. Correspondence: corresponding author has already graduated from Lincoln University. Institutional email is no longer accessible.
3. A range of protein content is provided in lines 54-62.
4. Line 90: "were" has been replaced by "were purchased"
5. Line 79: the authors used unpasteurized frozen deer whole milk. How authors manage the initial bacterial contamination in milk? How did they distinguish peptides originating from gastric duodenal and peptides initially present in milk due to proteolysis by bacteria. Similarly, generally, the determination of antimicrobial activities were performed on sterile raw materials (sterilized by heat or membrane filtration) which are subsequently contaminated by different bacterial strains. In this context, how authors deal with the fact that unpasteurized milk was used?
The use of unpasteurized milk to produce peptides is standard and has been used in milk from different species [please see Mudgil et al. (2018), Characterization and identification of novel antidiabetic and anti-obesity peptides from camel milk protein hydrolysates. Food Chemistry, 259, 46-54; and Hodgkinson et al. (2019). Gastric digestion of bovine and goat milk: Peptides derived from simulated conditions of infant digestion. Food Chemistry, 276, 619-625.]. Unpasteurized milk is used to avoid interferences caused by thermal treatment that causes protein changes prior to hydrolysis and can add confounding effects. This is in line with many previous studies and some references were given above. Fresh milk collection was from research farms and collected under strict hygienic conditions. Further, freezing causes substantial reduction and control of microorganisms (Katsiari et al., 2002; Food Chemistry, 77, 413-420, Manufacture of yoghurt from stored frozen sheep’s milk) and the whey was filtered using 0.45 µm DVPP filter as stated in L129 prior to any chromatography work.
The study targeted Lf after that protein was purified using chromatography. It is unlikely that any bacteria was in the starting material due to the reasons above. Any that was present would have been removed during purification and the starting material for hydrolysis (i.e. Lf) would be obtained free from bacteria. Furthermore, any hydrolysates that were present would have beeen removed during the purification steps to obtain Lf prior to hydrolysis.
The peptides were sterilized by the heating at 80 degrees for 15 min (as stated in L159), thus there is no contamination from the peptides.
6. Replaced ml/L by mL/L throughout the manuscript.
7. Replaced "defatted" by "skimmed" .
8. Bovine milk was used as reference material “a control group” line 108. The same processes were applied on milk to obtain bovine milk whey line 109.
9. Line 123: How authors discriminate proteolysis induced by digestion vs proteolysis in initial milk due to bacterial contamination since milk was not pasteurized?
Please our response above regarding proteolysis by bacteria. As you can see from the starting material there is no degradation in the Lf from either species.
10. Details were added regarding the preparation of SDS-PAGE (boiling step, add of beta mercaptoethanol, etc.) lines 168-182.
11. Line 142: Similar to my previous comments: How samples were sterilized for antimicrobial analysis.The whey was filtered using 0.45 µm DVPP filter as stated in L129. The study targeted Lf a protein that was purified using chromatography. It is unlikely that bacteria, if any, will be in the starting material. Furthermore, any hydrolysates, if any, will be removed during the purification steps to obtain Lf. The peptides were sterilized by the heating at 80 degrees for 15 min (as it was stated in L159), thus there is no contamination from the peptides.
12. Line 220: deleted.
13. Fig 2 and 3: protein bands from bovine milk are the reference for those from deer milk.
14. Table 1: bovine milk is a control and reference for deer milk.
Figures 4 and 5: bovine milk is a control and reference for deer milk.
Reviewer 2 Report
The authors wrote a manuscript to determine the bioactivities of individual whey proteins and hydrolysates from deer milk.
Please find my comments below:
Introduction
Please, increase the introduction including a paragraph focus on physio-chemical and quality parameters of Deer milk and their importance of cheesemaking.
Line 58-67 please, move these sentences for material and methods
Material and methods
Line 142 – 170 Separate the section in Bacterial Strains and Determination of antibacterial activity.
Results
Line 236-245 I do not understand this paragraph and the table, please rephrase this paragraph and change the layout of the table. In addition, please, explain better what are X and MIC in the table.
Line 250 3.3.2. Antibacterial activity of deer and cow Lf hydrolysates
I would like to understand why the authors used only E. coli ATCC 25922 in this essay?
Fig 4 and 5 please add the Standard deviation and change the background.
Discussion
Line-322 the authors evaluate the influence of antibiotic resistance in the strains studied? Which relevance this sentence has to work? Please explain better.
Conclusion
The antimicrobial conclusion is very poor, please improve it.
Author Response
The authors thank the reviewers for their useful comments and for their time reviewing the manuscript.
Response to reviewer 2 comments:
1. Physio-chemical and quality parameters of Deer milk and their importance in cheesemaking: lines 47-49, 51-57.
2. Line 58-67 moved to material and methods lines 95-103.
3. Separate the section in Bacterial Strains lines 170-190 and Determination of antibacterial activity lines192-197.
4. Rephrased paragraph 3.3.1. Table 1: NA: not available. MIC: Minimal Inhibitory Concentration (explained in Abbreviation section.)
5. Antibacterial activity of deer and bovineLf hydrolysates
Table 1 shows the inhibitory activities of deer Lf, bovine Lf, and their hydrolysates against three strains E. coli ATCC 25922, S. aureus ATCC 25923, and L. acidophilus ATCC 4356. The E. coli ATCC 25922 is the most susceptible bacteria to Lf and hydrolysates. The grow curves of E. coli ATCC 25922 in the presence of deer Lf, bovine Lf and their hydrolysates show significant differences compared with the bacteria incubated in broth which was the control group. Therefore, only the growth curves from E. coli ATCC 25922 are shown in the draft, but not S. aureus ATCC 25923 and L. acidophilus ATCC 4356.
6. No standard deviations are shown in Fig 4 and 5 because adding SD will make the figures messy and difficult to notice the clear trend of the growth curve.
7. Discussion
Lines 352-356. No antibiotic resistance in the strains was studied. It’s addressed that bacterial cell density can affect the Minimal Inhibitory Concentration.
8. Conclusion is revised.
Reviewer 3 Report
Brief summary:
Using ion exchange chromatography, the authors of the present study succeeded in isolating lactoferrin (and β-lactoglobulin), for the first time, from red deer milk. This lactoferrin was shown to be susceptible to hydrolysis by pepsin, trypsin, and chymotrypsin. Gastric and duodenal digested hydrolysates of deer milk-derived lactoferrin had a strong antibacterial effect on Escherichia coli ATCC 25922. In conclusion, deer milk was found to be capable of generating bioactive peptides that may beneficially influence human health by inhibiting various food-borne pathogenic bacteria.
Broad comments:
This is a reasonably sound study. The experiments performed by Wang and co-workers have been conducted rigorously, although sample sizes were not large enough to produce very robust results. The methods and reagents used have mostly been described in sufficient detail. The data presented in the manuscript support the conclusions reached.
Specific comments:
- I suggest that the gross (typical) chemical composition of red deer milk be given in the Introduction section.
- Please replace “78056 Da” with “78.1 kDa” in line 35.
- The lactoferrin content of bovine milk was reported by several authors to range between approximately 100 and 200 mg/L or 0.1 and 0.2 mg/mL (https://doi.org/3168/jds.2006-827, https://doi.org/10.3168/jds.2007-0689, https://doi.org/10.3168/jds.2008-1255). The corresponding value indicated in line 42 (i.e., 2.33 mg/mL) is an order of magnitude higher than those levels. Please discuss this issue.
- The sources of materials and instruments are supposed to be uniformly and properly identified (i.e., manufacturer’s name, city, country). Especially, the city is missing in several cases.
- Line 170: There is no “n” (number of generations) in the mathematical formula shown five lines above.
- Please define the abbreviations used in a specific figure in the title of that figures. I am not sure all potential readers know what GMP, Lf, Lp, etc. stand for.
- The number of observations (n) should be indicated in the footnote to Table 1, e.g., “Values are means ± SD based on 3 observations.”
- Similarly, I failed to notice the number of observations (n) in Figures 4 and 5.
- I suggest that the first paragraph of the Conclusion section be deleted because it is actually a summary. The second paragraph is well written, though.
Author Response
The authors thank the reviewers for their useful comments and for their time reviewing the manuscript.
Response to reviewer 3 comments:
- Chemical composition of red deer milk: line 32-34.
- Replaced “78056 Da” with “78.1 kDa” in line 35.
- Corrected units of Lf in deer and bovine milk in line 42 and line 43.
- The manufacturer and city of materials and instruments have been added.
- Deleted n=number of generations in Line 172.
- Abbreviations were added from Line 19-37.
- Values are means ± SD based on 3 observations in line 270.
- All the curves are the average of triplicate incubations in line 289 and 293.
- The first paragraph of the Conclusion section is deleted.
Round 2
Reviewer 1 Report
The authors responded effectively to comments indicated in the first round of manuscript review. Minor revision are requested, mainly regarding the quality of figures at line 302 and Figures 4 and 5 (decrease the size and remove gridlines of excel figures).
Author Response
The authors thank the reviewers for reviewing the revised manuscript.
Actions: Decreased the Excel figures' size and removed gridlines of bacterial growth curves.
Reviewer 2 Report
The authors followed the suggestions from the reviewers. I consider the reviews done.
Author Response
The authors thank the reviewers for reviewing the revised manuscript.